# Pre-Training Transformers for Fingerprinting to Improve Stress Prediction in fMRI

**Gony Rosenman**[*,1] **Itzik Malkiel**[*,2] **Ayam Greental**[1] **Talma Hendler**[1] **Lior Wolf**[2]

[1]Sagol School of Neuroscience, Tel Aviv University
[2]School of Computer Science, Tel Aviv University

**Editors:** Accepted for publication at MIDL 2023

## Abstract

We harness a Transformer-based model and a pre-training procedure for fingerprinting on fMRI data, to enhance the accuracy of stress predictions. Our model, called MetricFMRI, first optimizes a pixel-based reconstruction loss. In a second unsupervised training phase, a triplet loss is used to encourage fMRI sequences of the same subject to have closer representations, while sequences from different subjects are pushed away from each other. Finally, supervised learning is used for the target task, based on the learned representation. We evaluate the performance of our model and other alternatives and conclude that the triplet training for the fingerprinting task is key to the improved accuracy of our method for the task of stress prediction. To obtain insights regarding the learned model, gradient-based explainability techniques are used, indicating that sub-cortical brain regions that are known to play a central role in stress-related processes are highlighted by the model.

**Keywords:** fMRI,Transformers,Metric-Learning

## 1. Introduction

Stress is the silent pandemic of our times. The immediate psychobiological response to stressful events, the *acute stress response*, aids the organism to adapt to environmental challenges and regain homeostasis. It reorients attention towards potential threats and facilitates behavioral and somatic protective responsivity, mediated by activity in the sub-cortical threat circuit, Salience Cortical Network along with deactivation of regulation systems, namely, the Central Executive Network and Default Mode Networks (Hermans et al., 2014; Arnsten, 2009).

A major obstacle in combating the debilitating health consequences of stress is that individuals differ greatly in how it affects them. While some individuals may encounter acute stress without substantial costs, others exhibit non-adaptive responses, which induce chronicity and debilitating dysfunctions (Arnsten, 2009). Converging neurobehavioural evidence from animal and human studies delineates differential processes of non-adaptive acute stress response which could lead to chronicity (Van Oort et al., 2017; Chattarji et al., 2015). Considering these stress consequences, facilitating a computational paradigm for characterizing the neurobiology of stress in a non-invasive manner can potentially aid the diagnosis and treatment of millions worldwide every year (Yaribeygi et al., 2017; Godoy et al., 2018). However, the research of stress resilience, mental health despite adversity, has struggled to highlight meaningful neural and behavioral factors that explain why individuals vary in their coping with stress, and could guide prevention or early intervention in cases of

---

[*] Contributed equally.

poor adaptability (Kalisch et al., 2017). Our analytic approach provides a robust way to identify personal factors in fMRI resting-state scans performed before and after acute lab stress, which are mostly related to the unique features of individuals with respect to other subjects. Previous literature indicates that the "fMRI fingerprint" can produce insights related to various clinical and behavioral attributes.

Our proposed method, which leverages pre-training for fingerprinting is motivated by the above principles. The fingerprinting training specializes the model in a signal associated with behavioral attributes, which is also key to accurate stress prediction. Specifically, we employ 3D convolutional networks and a Transformer architecture to learn a fingerprinting function, by propagating a sequence of rs-fMRI through a model. The model, named MetricFMRI, outputs a vector that holds functional connectivity information and serves as a learning-based alternative to the traditional functional connectivity matrices computed with Pearson's correlation. The model optimizes a triplet loss function, that maximizes the similarity of vectors propagated from sequences of the same subject, while minimizing the similarity of vectors propagated from sequences of different subjects.

The MetricFMRI scheme operates as follows: we first train the model with self-supervision to solve an auto-encoding task of reconstruction of sequences of fMRI frames. Next, we optimize the model with the triplet training approach, using three scans that are denoted by "anchor", "positive" and "negative". The "anchor" and "positive" are related to the same subject, and the "negative" is sampled randomly from a different subject. The goal of this training phase is to teach the model to produce representations that are unique for different subjects. Following this training phase, the model provides representations that are a de-facto fingerprint of different individuals. We then leverage the pre-trained model for a target task, by fine-tuning its weights with supervision.

Another contribution is that we apply explainability methods to our transformer-based fMRI model. Very reassuringly, it is found that the model stress predictions are significantly affected by the Pallidum, Putamen, Thalamus, and Amygdala regions, which are known to play a central role in human stress processes. Finally, by applying the explainability technique to the pre-trained fingerprinting model, we observed that it highlights the sub-cortical and temporal-cortical regions. These regions are also associated with stress. Therefore, we hypothesize that the fingerprinting task reinforces the model to produce accurate representations for those regions, hence promoting the performance of stress prediction.

**Related Work**     A growing body of evidence suggests the existence of a "functional connectome fingerprint", a pattern unique to each brain that is acquired through the resting-state functional connectivity matrix (FC) of an individual brain. This matrix depicts the Pearson's correlation between every pair of parcels, based on the BOLD signal during a resting-state fMRI scan. The original concept was suggested by Finn et al. (2015), who used it to identify, with a high degree of accuracy, the resting-state fMRI scans that belong to the same individual brain. Fingerprinting was subsequently linked to various clinical and behavioral attributes. Van De Ville et al. (2021) explored the change in fingerprinting across time scales by enforcing a dynamic connectivity approach to produce connectivity matrices. Machine learning was introduced into the fingerprinting process as an alternative to vanilla correlation computation. Cai et al. (2021) use an autoencoder to reduce the shared components of the FC and increase inter-subject variability. Sarar et al. (2021) show the effectiveness of shallow feed-forward models in increasing the accuracy of fingerprint

prediction, for shorter lengths of resting-state scans. In a recent opinion piece, Finn and Rosenberg (2021) raise a concern that attempts to optimize the reliability of the fingerprint lead to a substantial amount of clinical and behavioral information being lost. It is proposed that instead of building metrics for promoting fingerprinting that are extremely accurate and reliable over time, the prediction of behavior should become the goal benchmark.

Other works have addressed the challenge of predicting stress from fMRI data at the level of the individual. Liu et al. (2021) use whole-brain functional connectivity data to predict individual stress during the COVID-19 pandemic. They found a critical role in communication between the limbic system and temporal lobe. Lee et al. (2021) focus on the task of predicting Psychophysiological Insomnia, a clinically important symptom of distress, using an individual-level machine learning approach. The input data consists of contrast images of cortical fMRI signal in multiple tasks. Other works (Long et al., 2014; Yang et al., 2021; Dopfel and Zhang, 2018; Weldon et al., 2015) that focus on predicting stressogenic symptoms at the individual fMRI level use a variety of machine learning approaches and input data, including animal data, reach similar conclusions about the contribution of limbic connectivity networks. The common pitfalls of such efforts are the sensitivity to smaller datasets and the challenge of binarizing a symptom that is spectral in nature.

## 2. Method

The MetricFMRI model employs a multi-phase training approach, where the model first pre-trains on a reconstruction and fingerprinting task with a metric-learning objective and then fine-tunes on a specific supervised task.

The MetricFMRI model adopts the TFF (Malkiel et al., 2022) architecture, which utilizes three components: (1) a 3D convolutional encoder $\mathcal{E}$, (2) a transformer network $\mathcal{T}$, and (3) a 3D convolutional decoder $\mathcal{D}$. The encoder is a 3D CNNs and operates on sequences of 3D volumetric data, and transforms them into sequences of 1D feature vectors, each vector corresponding to a specific fMRI frame. The transformer incorporates multi-head attention layers and operates on the output of the encoder network. The decoder is composed of 3D convolutional layers that map 1D vectors to 3D volumes of the same size as the input fMRI frames. The decoder operates on the output of the encoder network and is used only during the initial training phase to reconstruct the input.

**Pre-Training**  The MetricFMRI pre-training includes two steps. First, the encoder $\mathcal{E}$ and a decoder $\mathcal{D}$ are trained for reconstruction. This step allows the 3D convolutional encoder to learn an effective representation of fMRI data. Then, the decoder is removed and a transformer model is employed on top of the encoder, and the model is trained to optimize a metric-learning objective on triplets of fMRI sequences. The latter reinforces sequences of the same subject to have representations with similar directions, while sequences of different subjects are pushed away from each other.

Given an fMRI scan with $n$ frames denoted by $X := (x_1, ..., x_n)$, where each $x_i$ is a volumetric fMRI frame representing the acquired pulses and echoes in a given interval, $x_i \in R^{W \times H \times D}$ where $W, H, D$ are the width, height, and depth of the acquired data. We first normalize each frame using the voxel normalization approach, which separately z-score normalizes the values of each voxel across the time domain of a given scan. The voxel normalization emphasizes the relative activation of each voxel in a given interval while suppressing structural information. We denote the normalized representations of the entire

scan as $\hat{X} := (\hat{x}_1, ..., \hat{x}_n)$. Fig. 8 in the supplementary materials presents a frame along with its voxel normalization.

By extracting a sequence of frames with a length $w$, the frames are aggregated on the batch dimension. Then, the batches of frames are propagated through the encoder and the decoder network, which outputs data with the same dimension as the input frames. The encoder and decoder are trained for reconstruction, optimizing a dual-loss term objective composed of MSE and perceptual loss (Johnson et al., 2016).

In the second pre-training step, the decoder is removed, a transformer model is applied on top of the encoder, and the model is trained by sampling triplets of fMRI sequences and optimizing a metric learning objective. Each triplet is composed of three sequences of fMRI frames, $(x_a, x_p, x_n)$ where $x_a$, $x_p$, and $x_n$ are *anchor*, *positive* and *negative* samples, respectively. The anchor and positive sequences are sampled from the same scan, but without temporal overlap, and the negative is sampled from a scan of a different subject. Each sequence is composed of a window of $w$ frames. The sequences are grouped on the batch dimension and propagated through the encoder, which maps each frame into a vector. The vectors of each sequence are grouped into a unified sequence. The special classification (CLS) token is added to the beginning of each sequence and then the sequences are propagated through the transformer model. The transformer outputs embedding for each of the input vectors, representing each of the frames and the CLS in a latent space. The embedding of the CLS of each sequence are then propagated through a triplet loss objective.

Formally, the triplet Loss objective is given by:
$$\mathcal{L}_T = L(a, p, n) = \max(0, m + d(a, p) - d(a, n)), \tag{1}$$
where $m$ is a pre-defined margin, $a, p, n$ are the CLS embeddings of each of the three sequences:
$$a = [f(x_a)]_o, \ \ p = [f(x_p)]_o, \ \ n = [f(x_n)]_o, \tag{2}$$

where $0$ is the index of the CLS token in each sequence and $f$ is the encoder-transformer network that operates on each sequence:
$$f = \mathcal{T}\left[\mathcal{E}\left(\left(\hat{X}\right)_{si}^{si+w}\right)\right]_0 \tag{3}$$
and $d(u, v) = 1 - \frac{u \cdot v}{\|u\|\|v\|}$ is the cosine distance.

Using a triplet-loss objective, the model embeds fMRI sequences from the same subject with feature vectors pointing in similar directions, while sequences from different subjects are separated by a margin.

**Supervised Fine-Tuning** During fine-tuning, the encoder and transformer networks can be optimized to a specific supervised task, by adding a standard classification (regression) head on top of the embedding of the CLS token.

The fine-tuning objective can be expressed as $\mathcal{L}_{fine-tuning} = -\Sigma_{i=1}^{m} \mathcal{L}_{cce}\left(y_i, \mathcal{C}\left(\mathcal{T}\left[\mathcal{E}\left(\hat{x}_i^w\right)\right]_0\right)\right)$, where $\mathcal{C}$ is the classification (or regression) head, $x_i^w$ is a sequence of $w$ frames associated with the label $y_i \in \{1...c\}$ ($c$ is the number of classes), $m$ is the number of sequences in the train set, $\mathcal{L}_{cce}$ is the softmax function followed by a standard categorical cross-entropy loss, and $0$ is the index of the CLS token.

**Inference**    Given an fMRI scan $X$, we compute $\hat{X}$ and extract all sequences of length $w$ and stride $s$. The MetricFMRI inference operates as follows: $\text{MetricFMRI}_{\mathcal{I}} := \frac{\sum_{i=0}^{m} \mathcal{C}\left(z\left((\hat{X})_{si}^{si+w}\right)_0\right)}{m}$ where $m$ is the number of sequences for the given stride $s$ and $z(\hat{x}) = \mathcal{T}\left[\mathcal{E}\left(\hat{X}\right)\right]$.

To predict whether two samples $\hat{X}$ and $\hat{Y}$ are associated with the same subject, we calculate the cosine similairty between the embedding of the sequences by $s(\hat{x}, \hat{Y}) := 1 - d\left(\frac{\sum_{i=0}^{n} z(\hat{X})_i}{n}, \frac{\sum_{i=0}^{n} z(\hat{Y})_i}{n}\right)$ retrieving true if it is above a threshold $\tau$ and false otherwise.

## 3. Experiments

We evaluate MetricFMRI on the fingerprinting and stress prediction tasks.

**The data**    In this study, we use the Combat Pilots fMRI Scans dataset (CPS) collected at Souraski medical center, Tel Aviv, as part of the study "Neural Indications of Stress-Induced Mental Overload"[1]. Two groups of scanned subjects are included in the dataset: combat pilots and non-pilots. In total, the dataset contains 50 subjects, each scanned before and after acute-stress conditions. Resting-state fMRI activity was measured for each subject and its two scans (before and after the exposure to stress). In this study, we focus on a binary classification task for predicting whether a scan was taken after stress exposure. Stress was induced using a well-established stressful task(Dedovic et al., 2005). Structural and functional scans were performed in a 3.0 Tesla Siemens MRI system,with a twenty-channel head coil. Structural scans included a T1-weighted magnetization prepared rapid gradient echo (MPRAGE) (TR/TE = 1860/2.74 ms, flip angle = 80, voxel size 1.0x1.0x1.0 mm, FOV = 256×256 mm, slice thickness = 1 mm). Functional whole-brain scans were performed in an interleaved order, using a T2*-weighted gradient echo-planar imaging pulse sequence (TR/TE = 3000/35 ms, flip angle = 90°, voxel size 2.3x2.3x3.0 mm, FOV = 220×220 mm, slice thickness = 3 mm, 45 slices per volume). Raw DICOM data images were converted to NIFTI format and organized to conform to the 'Brain Imaging Data Structure' specifications (BIDS). Preprocessing was conducted using FMRIPREP version 1.5.863, a Nipype-based tool64. More details about the data can be found in the supplementary, Sec. B.

**The baselines**    We compare MetricFMRI with three state-of-the-art methods: Spatial-Temporal Graph Convolutional Networks for fMRI (ST-GCN) (Gadgil et al., 2020), DeepfMRI (Riaz et al., 2020), and Transformer Framework for fMRI (TFF) (Malkiel et al., 2022).

*Spatial-Temporal Graph Convolutional Networks for fMRI (ST-GCN)* is a technique that was recently evaluated on age and gender prediction from fMRI scans. Operating on resting-state fMRI volume data, this model transforms fMRI frames into vectors by parcellating sequences of the scans using a standard brain atlas. The vectors are then normalized and fed into the ST-GCN architecture, which has shown efficacy in learning from graph-structured time series (Yu et al., 2017).

*DeepfMRI* is a recent model that operates on fMRI sequences and can be trained with various fMRI prediction tasks. It parcellates the volumes using a brain atlas, outputting a single vector for each fMRI frame. The vectors are then fed into a neural network that learns to predict the connectivity matrix of the given input. The network architecture is composed of a sequence of 1D convolutional layers followed by fully connected layers. The matrix is then propagated through an additional network that outputs a prediction.

---

[1] https://www.clincosm.com/trial/healthy-stress-psychological-tel-aviv-and-cognitive-load-induction

Table 1: Stress prediction results on the CPS dataset.

| Model | BAC | Acc. | AUC |
|---|---|---|---|
| MetricFMRI | **78.84±9.74** | **78.84±9.74** | **81.11±10.08** |
| ST-GCN | 62.5±5.41 | 62.5±5.41 | 63.12±7.13 |
| Deep-FMRI | 56.25±3.44 | 56.25±3.44 | 58.37±5.8 |
| TFF | 52.3±1.88 | 52.3±1.88 | 43.71±6.21 |

*Transformer framework for fMRI (TFF)* employs a two-phase training approach and leverages 3D convolutional networks and a Transformer architecture. TFF applies self-supervised training to a collection of fMRI scans by optimizing the model to reconstruct 3D volume data. In the second phase, the model is fine-tuned for specific tasks with supervision. In our work, we adopt the same architecture and add a fingerprinting learning phase.

While similar in architecture, our model is much more efficient than TFF, which has two pre-training phases with a reconstruction loss: the first one, similar to our work, employs an encoder-decoder architecture, and the 2nd pretrains the transformer as encoder-transformer-decoder. The second phase of TFF requires days of training on decent hardware due to the computational complexity entailed by the encoder-transformer-decoder architecture. In MetricFMRI, we omit this phase and propose a much more efficient encoder-transformer fingerprinting-based training. More details can be found in the supplementary Sec. E

We note that the baseline methods were previously evaluated on datasets of larger size: ST-GCN, DeepfMRI, and TFF were evaluated on datasets with ∼1000, ∼700, and ∼200-1000 subjects, respectively. Our study predicts stress using the CPS dataset, which is an order of magnitude smaller and matches in size many of the current fMRI studies.

**Implementation details**   MetricFMRI utilizes the AdamW (Loshchilov and Hutter, 2017) optimizer, with a weight decay of 1e-7 during the first pre-training phase where the encoder and decoder are trained for reconstruction, without the transformer, and 0.01 during the second pre-training phase of the triplet objective as well as the fine tuning. The window size is set to $w = 30$ across all experiments, with a stride of $s = 7$. The encoder architecture imposes a bottleneck layer of size $d = 2640$. In our experiments, all MetricFMRI models were trained by using a single V100 GPU card. the two pre-training phases and the fine-tuning took 10, 15, and 25 epochs respectively, accumulating a total of approximately 30 hours of training. Weights are initialized with pytorch's 'kaiming_uniform' initialization.

**fMRI stress prediction results**   Table 1 depicts the performance of all models evaluated on the stress prediction task, reporting balanced accuracy (BAC), accuracy (Acc.) and area under the curve (AUC). In this task and dataset, we formulate the stress prediction task as a binary classification, by predicting high-stress or no stress. The performance of all models is reported for a K-fold cross-validation scheme, with $k = 5$ and the same splits, except for TFF, for which, due to its high computational cost (see Sec. 3), only the first split was used.

As can be seen, MetricFMRI outperforms all other alternatives by a sizable margin. specifically, MetricFMRI outperforms DeepFMRI by ∼20 points of accuracy, and by ∼21 in the AUROC metric. Compared to ST-GCN, we observe an improvement of ∼14 and 12 points in accuracy and AUROC. Compared to TFF, we observe that MetricFMRI improves by larger margins. This can be attributed perhaps to some overfitting that occurred in the pre-trained TFF model, indicating that solely pre-training on reconstruction can produce a suboptimal performance for relatively small datasets containing few tens of subjects.

Table 2: CPS dataset fingerprinting results.

| Model | BAC | Acc. | AUC |
|---|---|---|---|
| MetricFMRI | **82.95±4.09** | **82.95±4.09** | **86.39±6.32** |
| TFF | 51.22±1.1 | 51.22±1.1 | 53.81±1.42 |
| ST-GCN | 53.6±0.7 | 53.6±0.7 | 53.8±2.74 |
| Deep-FMRI | 52.49±1.64 | 52.49±1.64 | 53.1±1.4 |
| Pearson corr. | 55.32±3.08 | 55.32±3.08 | 57.73±5.41 |

Table 3: Ablation study results.

| Model | AUC |
|---|---|
| (i) w/o pre-training | 50.76±1.18 |
| (ii) w/o triplet loss | 51.21±0.94 |
| (iii) w/o margin | 53.34±1.66 |
| (iv) $\mathcal{E}$-$\mathcal{T}$-$\mathcal{D}$ | 51.12±0.45 |
| (v) w/o reconstruction | 51.05±1.16 |
| (vi) pair loss | 55.0±2.62 |
| Full method | **81.11±10.08** |

**fMRI fingerprinting results** We formulate the fingerprinting task as a binary classification task, by building a dataset of pairs of fMRI scans and assigning each pair with a binary label, indicating if they are related to the same subject. For this task, we build upon the CPS dataset described above.

We evaluate the performance of all models on the fingerprinting task. Each model was trained with a five-fold cross-validation scheme. Mean scores and standard deviations are reported across all five evaluations. We also compare with Finn et al. (2015), who employ Pearson correlation for fingerprinting.

Since MetricFMRI pre-trains for fingerprinting, we report its performance without additional fine-tuning. Given a pair of samples, we classify the pair as negative or positive by propagating the pair through the MetricFMRI model and calculating the cosine similarity score between their representations (see Sec. 2). If the score is higher than a threshold $\tau$, we set the pair as positive, otherwise negative. $\tau$ was set to 0.9 by computing the optimal threshold that separates positive and negative pairs on the validation set.

As can be seen in Table 2, MetricFMRI outperforms all other alternatives by approximately 30 percent for the accuracy metric and 27 percent for the AUROC metric. We attribute the superiority of MetricFMRI over the ST-GCN and Deep-FMRI baselines to its architecture and pre-training procedures that optimize the representations under a well-defined metric (see ablation study). The TFF model seems to struggle in this task and dataset, which we attribute to some level of overfitting in the pretrained model due to the relatively smaller size of this dataset (containing 50 subjects), while TFF was mostly evaluated on datasets with roughly one-order-of-magnitude more scans.

**Ablation study** We conduct an ablation analysis to showcase the importance of each component in MetricFMRI and report the performance in Tab.3. The following variants are considered: (i) MetricFMRI without pre-training. In this variant, we apply the fine-tuning procedure on a randomly initialized MetricFMRI model. (ii) MetricFMRI without the pre-training with the triplet objective. Here we only pre-train the MetricFMRI for reconstruction. (iii) triplet loss without a pre-defined margin - we train the MetricFMRI model without the margin $m$ on Eq. 1. In this variant, the anchor-positive and anchor-negative pairs are pushed to a cosine similarity score of 1 and $-1$, respectively. (iv) pre-training $\mathcal{E}$-$\mathcal{T}$-$\mathcal{D}$ for reconstruction (instead of $\mathcal{E}$-$\mathcal{D}$ and then $\mathcal{E}$-$\mathcal{T}$ with triplet training), then applying the proposed triplet training (a single split is reported, due to time constraints). (v) omitting the training for reconstruction and directly training $\mathcal{E}$-$\mathcal{T}$ with the proposed triplet paradigm. (vi) replacing the triplet objective with a pair-based loss, that encourages positive (negative) pairs to have a cosine of 1 (-1).

The results, shown in Tab. 3, indicate that it is crucial to apply the pre-training in the way it is done in MetricFMRI and that the triplet loss, with the margin-based objective, is highly beneficial for convergence. Note that the ablation variants yield smaller variances, since their performance is always around an AUC of 50%, while the full method generalizes much better on average, with some splits demonstrating better accuracy than others. Ablations (iv) and (v) imply that it is highly important to train the Transformer model on top of vector sequences with meaningful representation. Variant (vi) implies that the triplet loss is helpful in avoiding overfitting that can be caused by a more restrictive loss, which can be even more important in small datasets, such as those one finds in fMRI studies.

**Explaining MetricFMRI predictions** We have employed an explanation technique to the fine-tuned MetricFMRI model that was trained for stress prediction and analyzed the brain regions that affected the model the most during inference. We leverage a standard saliency map technique based on gradients (Simonyan et al., 2013; Sundararajan et al., 2017), also known as "vanilla-gradients", applied to the input fMRI frames. Specifically, for each fMRI sample in the test, we calculate the gradients on the input fMRI frames w.r.t. the dimension in the logit vector that is associated with the stress class. We calculate gradients for every frame in the sequence and across all sequences classified as stressful in the test set. The absolute values of the gradients across all frames are then averaged, resulting in one gradient volume with the same dimension as a single fMRI frame. The volume is parcellated into regions, and each region is assigned a score that is the average value of its voxels. Each score estimates the importance of the region w.r.t. the model decision since a higher score means gradients with a bigger absolute value that can strengthen the final model prediction for stress. Finally, the regions are sorted in descending order according to their score.

We observe that the highest scoring regions are located in the sub-cortex and include the pallidum, putamen, thalamus, and amygdala. This implies that the above regions significantly affect the model's stress predictions. Interestingly, these regions are found in multiple studies to have a central role in human stress-related processes (Zhang et al., 2019; Hartogsveld et al., 2022; Herrmann et al., 2020; Maron-Katz et al., 2016; Zhang et al., 2020).

We further apply the explainability technique to the pre-trained MetricFMRI model that was trained for fingerprinting, by calculating the gradients w.r.t cosine similarity score between a population of pairs of sequences associated with different subjects. Then, we infer a score for each region as described above. The obtained scores estimate the importance of each region w.r.t. the fingerprinting prediction. We observe that sub-cortical and temporal regions are associated with the highest scores, indicating that the fingerprinting training reinforces the model to specialize in those regions, that are also known to be correlated with stress. More details about the explainability method can be found in appendix A.

## 4. Summary

We present MetricFMRI and show that pre-training on fingerprinting can be beneficial for stress prediction. MetricFMRI leverages a 3D encoder-decoder and a transformer architecture, and pre-trains to minimize both reconstruction loss and a metric learning objective that is based on triplets of fMRI sequences. The pre-training, in which the model learns to produce representations for fMRI scans that pushes sequences of fMRI frames of the same subject closer while pushing away those of other subjects, is found to be crucial for improved performance. MetricFMRI is a general model which can be used for various other fMRI prediction tasks.

**Acknowledgements**

This project has received funding from the European Union's Horizon 2020 research and innovation programme under grant agreement No. 777084 (DynaMORE project), and No. 945539 (Human Brain Project SGA3), and also from the ISRAEL SCIENCE FOUNDATION (grant No. 2923/20) within the Israel Precision Medicine Partnership program. It was also supported by a grant from the Tel Aviv University Center for AI and Data Science (TAD).

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

# Supplementary Appendices

## Appendix A. MetricFMRI explainability

To that end we present an Explainability pipeline for MetricFMRI (EMF) capable of explaining the decision making process at a Spatio-temporal level. We describe the explainability technique under the context of a classification task; the regression tasks are analogous. EMF leverages a combination of parcellation with gradient-based saliency maps calculated on the acquired fMRI data. The saliency maps are calculated with respect to the specific class in the prediction head of the fine-tuned MetricFMRI model.

Given an fMRI scan, EMF splits the scan into sequences of size $w$, propagates each sequence separately through the fine-tuned MetricFMRI model, and calculates the gradients on the voxel normalized volumes, for each frame and sequence. The gradients are calculated with respect to a specific dimension in the logit vector (in classification tasks, this dimension represents the predicted score for a specific class). Formally, to explain the model decision for predicting the $k$th class with regards to a sequence from time $t$ to time $t + w$ for subject $s$, we denote the $k$th dimension of the logit vector by $p_k^s$

$$p_k^s = \gamma \left( \left[ \tau \left( \epsilon \left( \{ x_{s'}^t, x_s^{t+1} \dots x_s^{t+w} \} \right) \right) \right]_0 \right)_k \tag{4}$$

By calculating the gradients on the input data with respect to $p_k^s$, we get a saliency map, which is a volumetric data of the shape of the fMRI frames, that holds voxel-level information which dictates the contribution of each voxel to the final decision of the model. Enhancing the voxels associated with positive gradient values would strengthen the value of $p_k^s$. More specifically, in the case of a classification task, enhancing the voxels associated with positive gradients would encourage the model to raise its confidence for predicting the $k$th class. The operation of calculating the gradients of a certain logit element with respect to the input can be formulated by:

$$G := \frac{\partial p_k}{\partial x} = \frac{\partial \gamma \left( \left[ \tau \left( \epsilon \left( \{ x_s^t, x_s^{t+1} \dots x_s^{t+w} \} \right) \right) \right]_{0 \, k} \right)}{\partial x_{ij,k}^t} \in \Re^{W \cdot H \cdot D \cdot T} \tag{5}$$

## A.1. From saliency maps to ROIs

At this point, we attained a tensor of the same shape as the input (volumes x time frames), with each voxel representing the sensitivity of the model's decision to a small change in the value of the voxel. The next step is to map the raw saliency maps onto spatially defined brain regions and enable a more meta-analytic examination of the data. To this end, we use a combination of the cortical and sub-cortical Harvard-Oxford brain atlas, summing to a total of 108 brain regions of interest (ROIs). The mapping is applied to the raw saliency maps resulting in a time series of gradient aggregation per ROI.

The final Decision Explanation Graph (DEG) is created by summing the gradients over the time dimension, revealing the ROIs that throughout the sequence were contributing most dominantly to the decision, and also calculating the Pearson's correlation of each ROI's gradient time series with the other ROIs, revealing temporally correlated contributions of other ROIs.

The motivation behind DEG computations, including the correlation between ROIs, stems from architectural choices that adhere to align with existing neuro-scientific paradigms. The transformation of gradients into regions and the framing of gradient correlations can simplify the process for neuroscience researchers to adopt our explainability technique, compare it to others and extract meaningful insights.

The DEG pipeline can be formally created by the following steps: 1. Given a sequence of fMRI pre-processed according to MetricFMRI pipeline

$$\left\{ x_{i,j,k}^t, x_{i,j,k}^{t+1} \dots x_{i,j,k}^{t+w} \right\} \tag{6}$$

2. Propagate the sequence through a MetricFMRI model after it was trained on a classification task to completion. Attain class probabilities vector p:

$$p_k^s = \gamma \left( \left[ \tau \left( \epsilon \left( \{ x_{s'}^t, x_s^{t+1} \dots x_s^{t+w} \} \right) \right) \right]_0 \right)_k \tag{7}$$

3. Calculate the gradients on the probability of the true class with respect to the input. Attain the saliency map G:

$$G := \frac{\partial p_k}{\partial x} = \frac{\partial \gamma \left( \left[ \tau \left( \epsilon \left( \{ x_s^t, x_s^{t+1} \ldots x_s^{t+w} \} \right) \right) \right]_0 k \right)}{\partial x_{ij,k}^t} \in \Re^{W \cdot H \cdot D \cdot T} \tag{8}$$

4. Parcellate G using a combined cortical/sub-cortical brain atlas, attain ROI time series table RT (in this study we used a combination of cortical and sub-cortical harvard oxford atlases)

$$RT := \text{ Parcellation } (G) \in \Re^{\text{num ROIs} \cdot T} \tag{9}$$

5. Sum RT over the time dimension to attain an overall ROI contribution score (ORC)

$$\text{ORC} := \sum_t^{t+w} RT \in \Re^{\text{mum ROIs}} \tag{10}$$

6. Extract correlation matrix from RT, explaining the Pearson's correlation between every pair of ROI's gradients in the sequence.

$$M := \text{corr}(RT) \in \Re^{\text{num ROIs.num ROIs}} \tag{11}$$

7. Conclude DEG by plotting the highest scoring ROIs from ORC alongside the ROIs that are the most correlated with them.

It is important to notice that the computation of DEG is done per sequence, so there are multiple DEGs per subject. The final graph is the accumulation over a certain group of subjects.

## A.2. EMF for embeddings

Until now, we have seen how to generate DEGs for MetricFMRI classification models, i.e., how to explain the model at the classification decision level. In the case we want to explain decisions at the embedding level, as in the case of explaining the fingerprinting task, we need to slightly modify the process. During the fingerprinting task, our input is two sequences of the anchor/positive (negative) pair, and the output is the cosine similarity of the anchor/positive (negative) pairs. By computing the gradients on the cosine similarity with respect to the input, we can measure which ROIs contribute to the similarity or dissimilarity of the embeddings. It is important to note that in that case, negative gradients convey contribution to difference and positive gradients convey contribution to similarity.

## A.3. Effective ROI contribution (popularity index)

Another way to summarize at a higher level the findings of EMF is through the Effective ROI Contribution score (ERC). Effective contribution is defined as the sum of individual ROI gradients over the scan plus the product of the specific ROI and the correlation with other ROIs. ERC quantifies the degree to which an ROI contributed to the decision weighted with respect to how much that contribution correlated with other ROIs, similar to the popularity index in graph theory. We can formulate ERC mathematically in the following way:

$$ERC_i = ORC_i + \sum_{j \neq i}^{\text{num ROIs}} ORC_i \cdot M_{i,j} \tag{12}$$

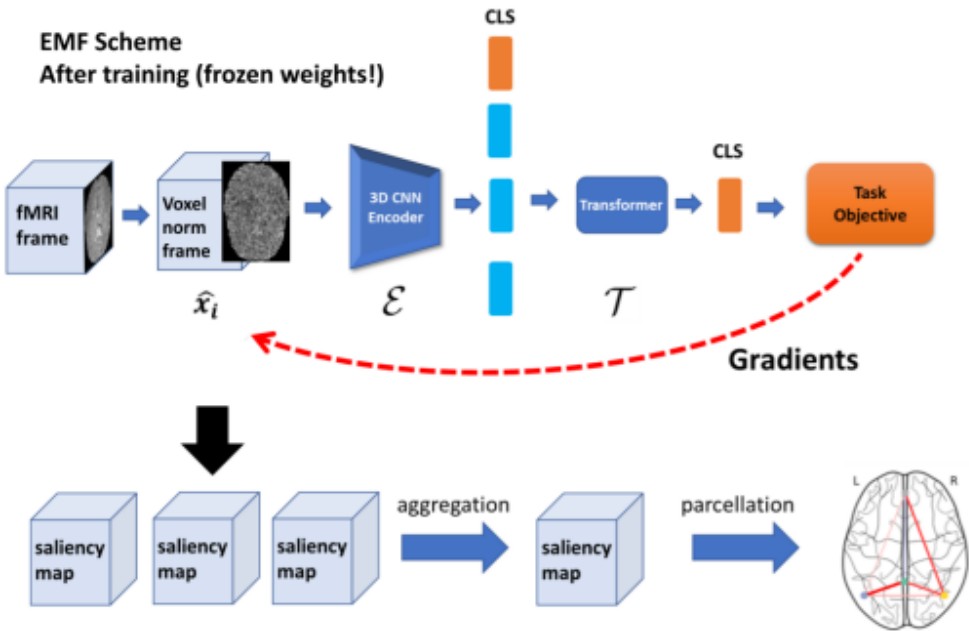

Figure 1: An illustration of the gradient explainability method.

ERC is presented over a smaller parcellation scheme that is a combination of the YEO 7 networks and the harvard-oxford sub-cortical atlas.

### A.4. From sample level DEGs to insights in neuropsychiatry

The end goal of MetricFMRI is to facilitate insights in the field of neuropsychiatry. By training an end-to-end deep learning model on a specific task, we can identify patterns at high Spatio-temporal resolution and then use DEGs to visualize similarities among sub-populations. In order to do that we show different usages of the EMF pipeline that put the emphasis on grouping DEGs under some parameter and discovering patterns that the model identified as shared among the group. In other words, EMF can generate DEGs per sequences, but some sequences have similar DEGs that may hint at a similar underlying neurological process.

First we group the DEGs that originate in Pilots and compare them to DEGs that originate in non-pilots:

A clear pattern of model sensitivity is centered in the left hemisphere region for both groups, with shared anti-correlations to deep limbic regions and distinct correlations to the right hemisphere. Anti-correlation can be interpreted as a regulatory or inhibitory relationship between regions. Statistically significant differences between the groups were measured in auditory processing regions, with clear opposite effects of the left central opercular cortex and left Hessle's gyrus for the pilot group. This distinction might hint at variations in brain structure and function that are the result of life-long training for the pilot group, their tolerance, and coping with stress as manifested in auditory processing and limbic connections.

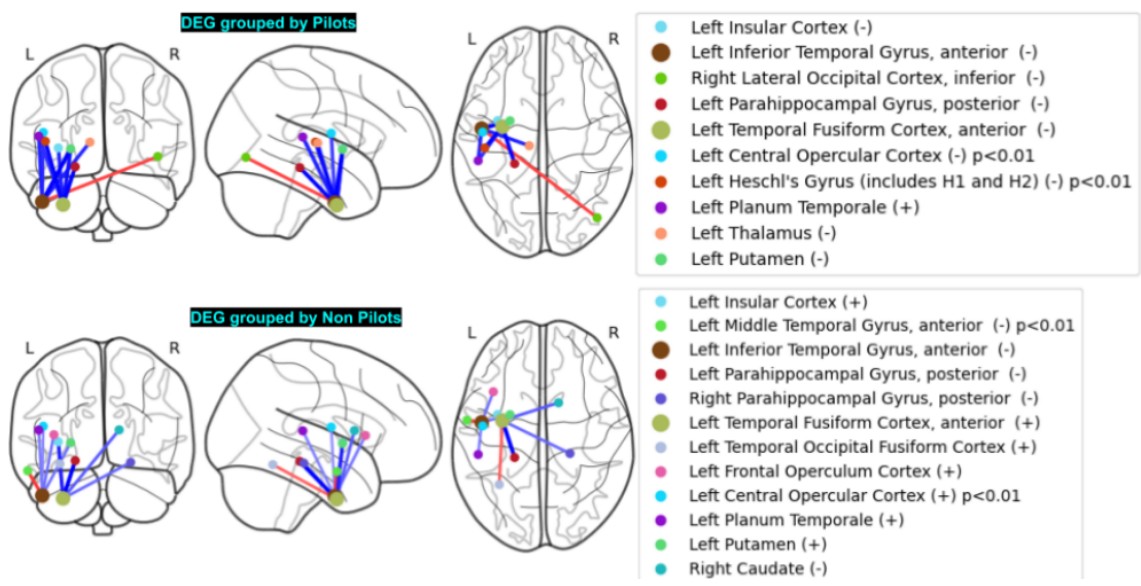

Figure 2: Comparing the mean DEG of the pilot group with that of the non-pilot group. Effects of life-long training can be associated with statistically significant correlative gradients between the Fusiform cortex and Heschel's gyrus in the pilot group.

Next, we group the DEGs that originate in subjects that exhibit impaired performance during the stressful aBAT task and compare them to DEGs that originate in subjects that exhibit improved performance:

Both groups demonstrate sensitivity to left inferior temporal regions, with anti-correlated connections to left insular and left temporal fusiform cortical regions, that can be interpreted as regulatory communication. It is interesting to note statistically significant differences among the groups - the impaired group shows sensitivity to the right thalamus while the improved group is to the left thalamus, and the gradient sign is opposite, meaning in the impaired group the model interpreted right thalamus activity as a promotor of stress. This overall distinction at the basal ganglia level hints at learning under stress coping mechanism that is altered for some subjects hence the impaired performance.

Another perspective is offered by the ERC score that is analogous to the popularity index in graph analysis.

These findings stand out from traditional contrast analysis in that it is the outcome of a machine learning model, and the gradients express the process embedded in the learning of the model and its learned sensitivity. There is still a lot to investigate concerning the gradient "signal", but the results so far offer a positive approximation of what this method has to offer. More experiments will increase the certainty of the mechanism that is forming under EMF and its ability to create valid explanations.

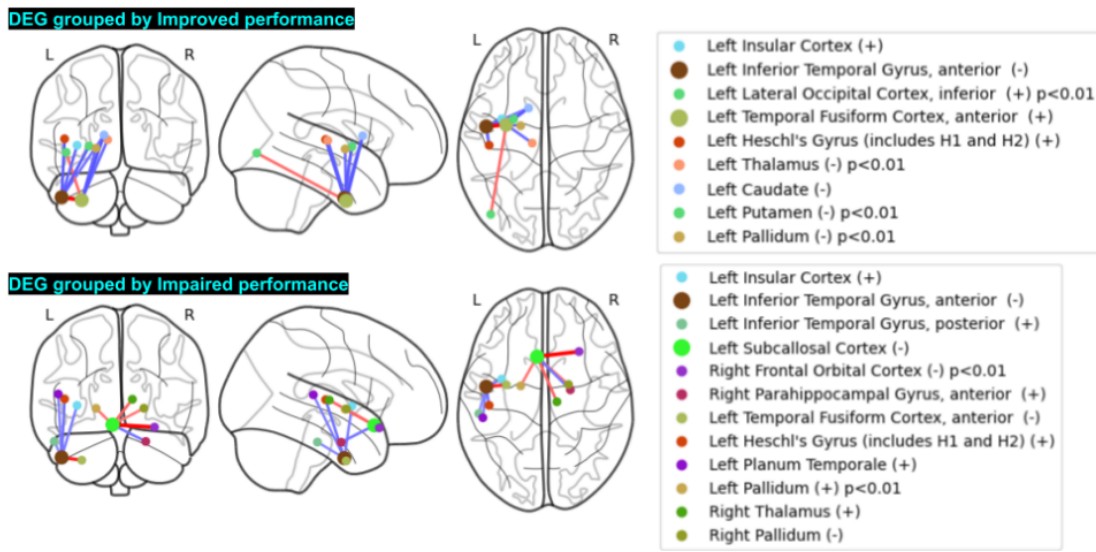

Figure 3: Comparing mean DEG of the impaired performance group with that of the improved performance. Statistically significant changes were measured in the Basal Ganglia ROIs, hinting at a different learning mechanism expressed in the gradients.

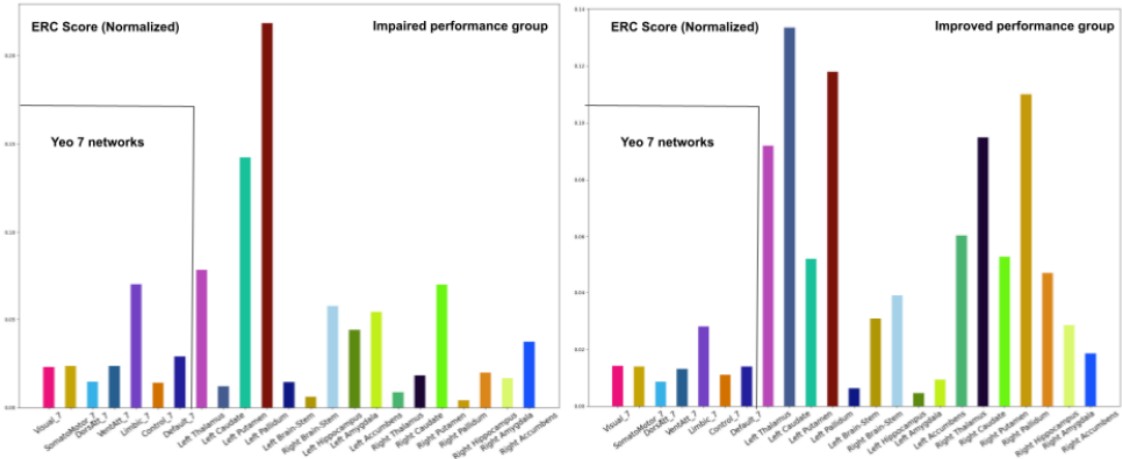

Figure 4: ERC scores of the improved performance group compared to the impaired performance. The high popularity of the Basal Ganglia regions hints at a system-level integration in the improved performance group.

### A.5. Fingerprinting - understanding the graphs

1. ROIs denoted by large nodes are the nodes with the highest mean absolute gradient value, and the small nodes are the ROIs with the highest correlative gradients to the large ROIs. for graphical convenience, only the two largest ROIs are shown and their top 90there are more than 5 in the top 902. Positive (negative) sign implies contribution to pair similarity (dissimilarity). an ROI with a positive gradient value is an ROI that slight perturbations to its BOLD signal resulted in the model increasing its confidence about pair similarity (higher cosine similarity score). 3. Red (blue) line implies gradient correlation (anti-correlation). A pair of ROIs have a large correlative gradients value if 5-fold exhibited temporally correlated gradients. The model was sensitive to both ROIs in a temporarily coherent way. 4. P values refer to the two-sided T-test performed between the value within the cluster compared to the value in the total population.

### A.6. Fingerprinting - clustering analysis

We start by showing the 4 most distinct DEG patterns that were discovered using the DBSCAN clustering algorithm:

We can see that the emerging clusters are distinguishable, yet all exhibit cross-lateral connections and sensitivity to the temporal and lymbic regions. Specifically, the most frequent pattern, DEG #1, shows sensitivity to the left subcallosal cortex and right inferior temporal gyrus. The effect these regions had is negative, which means it contributed to pair dissimilarity. DEG 2# exhibits parahippocampal importance that is centered in the right hemisphere but is also connected to the left. It contributed to the similarity of pairs. DEGs 3# and 4# are the least frequent yet they exhibit interesting patterns, both related to temporal and limbic regions but slightly different with connections to cross lateral temporal regions and fusiform cortical regions respectively.

Next we compare the ERC scores computed and averaged across the 4 clusters.

Score of each ROI/large-scale network in the combined YEO7 and sub-cortical Harvard oxford atlas. The limbic network was highly contributing throughout the entire population. The ERC scores further highlight the contribution of the limbic network, as it is projected when examining network-level parcellation combined with sub-cortical parcellations. The exact ROIs that form the yeo7 definition of the limbic network is appended to this study in the appendix section.

In total, we can see clear unique patterns that point out the model's broad scope of pattern identification, with high significance.

## Appendix B. More details about the dataset

In this study, we use the Combat Pilots fMRI Scans dataset (CPS) collected at Souraski medical center, Tel Aviv, as part of the study 'Neural Indications of Stress-Induced Mental Overload'. Two male groups (age $31.37 \pm 7.1$ years) of scanned subjects are included in the dataset. individuals with experience as combat pilots (n=20) and without such experience (n=30). Participants were scanned for fMRI during resting-state before and after participating in a stressful task (altogether 4 resting-state scans). Stress was induced using a multi-tasking procedure based on The Boundary Avoidance Task (BAT)(Faller et al., 2016)

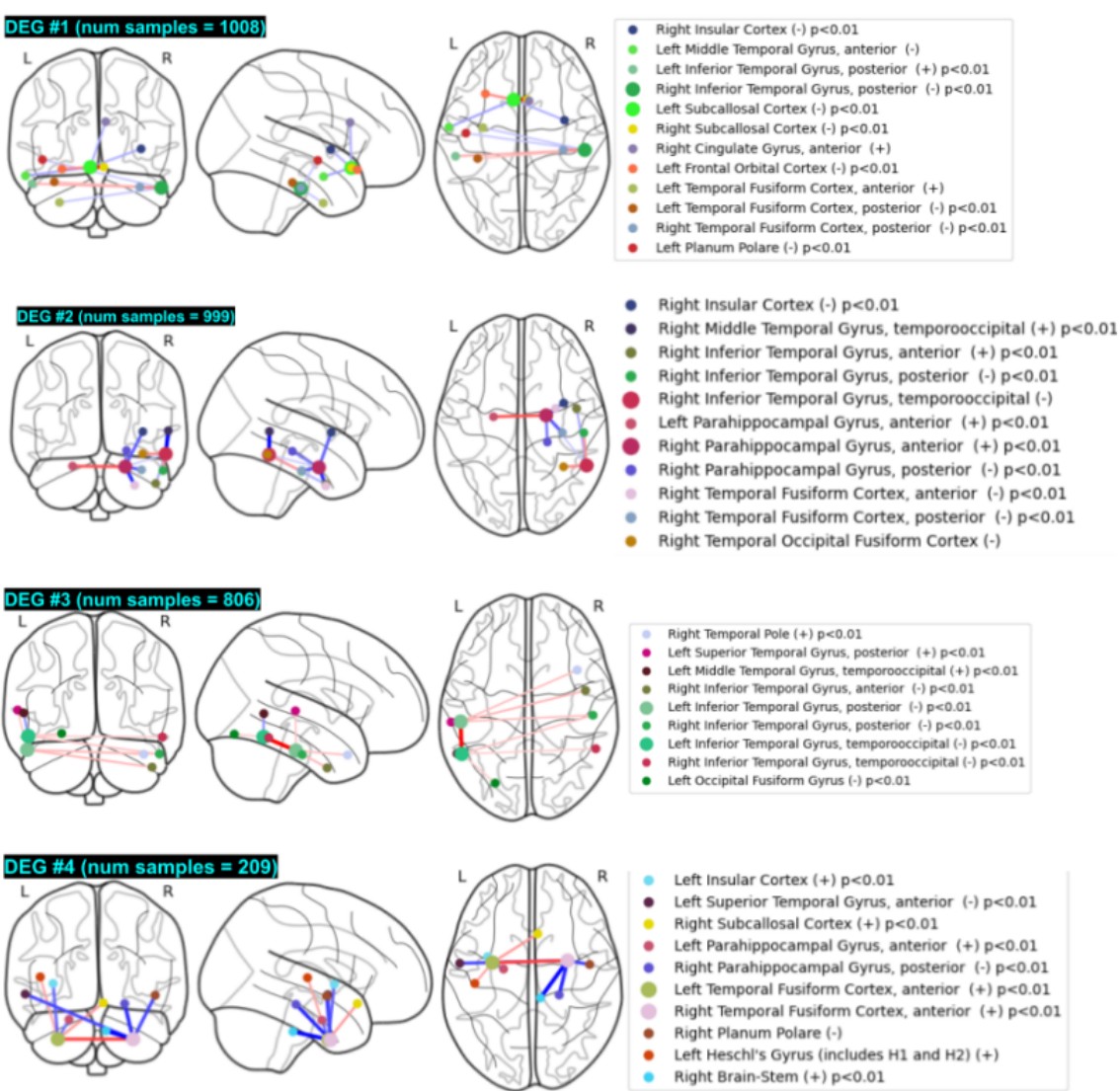

Figure 5: Most distinct subtypes of DEGs for the fingerprinting task, as captured with DBSCAN clustering algorithm. The mean of each cluster is shown.

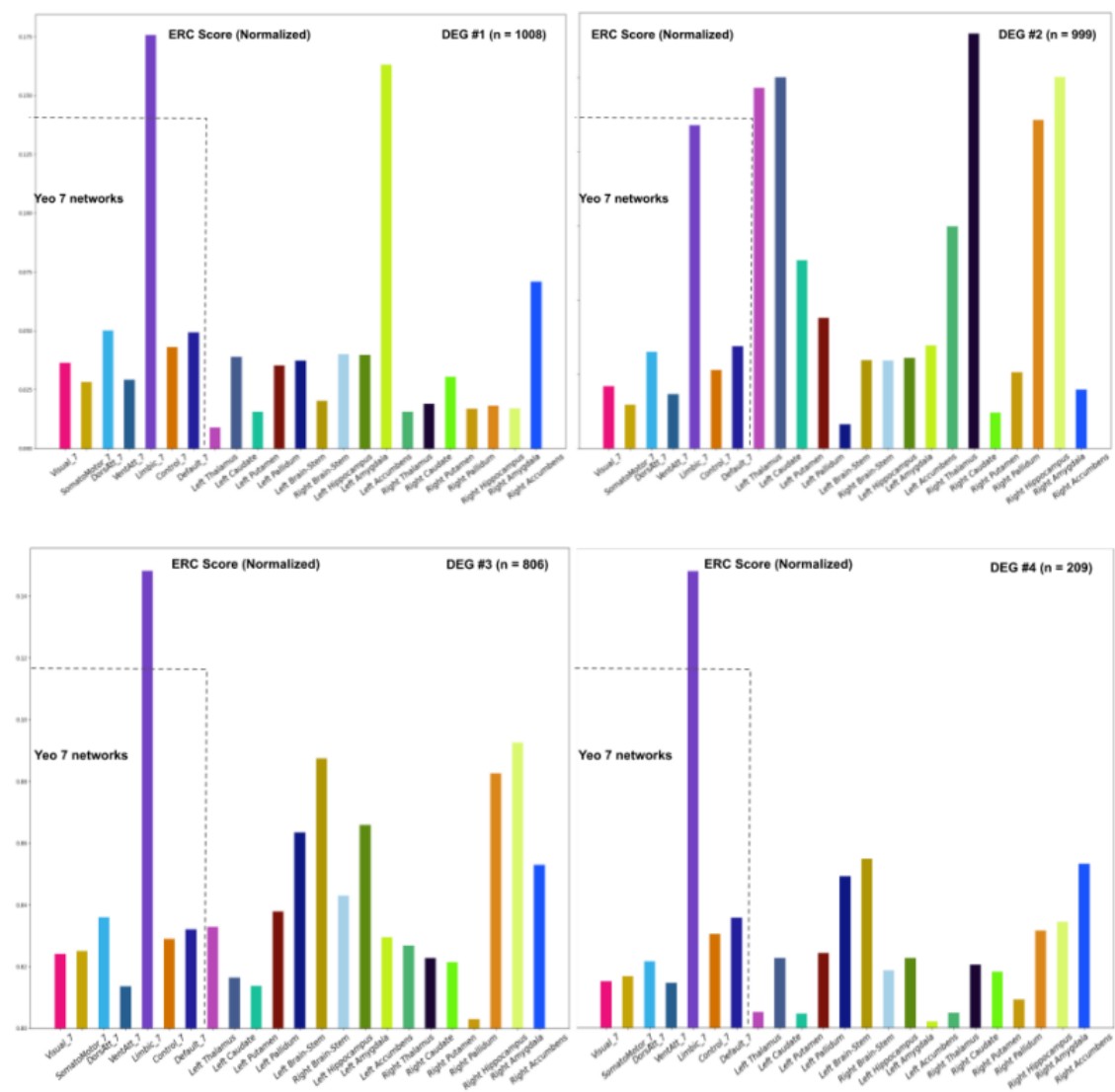

Figure 6: ERC score of each ROI/large-scale network in the combined YEO7 and sub-cortical Harvard oxford atlas. The limbic network was highly contributing throughout the entire population.

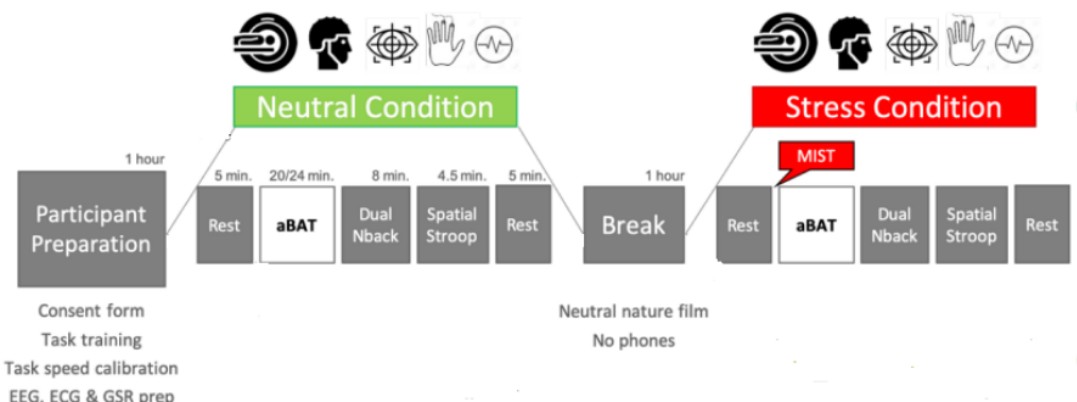

Figure 7: Experimental design. Each participant underwent a total of four resting-state sessions. Pre-task (neutral condition), post-task (neutral condition), pre-task (stress condition), post-task (stress condition). Only post-task (stress condition) is treated as a stressful state in our formulation.

which simulates a high cognitive workload combined with two parallel executive tasks, the N-back (Kirchner, 1958), and Spatial Stroop(Hilbert et al., 2014). Altogether this simulated gradually increased cognitive load into the original BAT, forming a new task named advanced BAT (aBAT). During one of the two stressful task sessions, another component of social evaluative stress was induced using the Montreal Imaging Stress Task (MIST)(Dedovic et al., 2005) which is derived from the Terier Mental Challenge(Kirschbaum et al., 1991), defining a high psychological stress condition. High and low-stress induction capabilities of the aBAT and the aBAT+MIST were verified beforehand with separate participants through a rise in cortisol levels. In our formulation, only the MIST-boosted post-stress sessions were considered stressful for our model's prediction.

## Appendix C. more details about the fingerprinting task

In Tab. 1 we report the performance of all models on the binary stress prediction task. In this task, we trained each model in a five-fold cross-validation scheme, and report the mean scores of the 5 models. Each model received the same five folds, preventing biases caused by random splits. All models shared the same training objective, which is a binary cross-entropy loss function. This objective treats fMRI sequences that were acquired during the Post High stress resting-state scan as positive stressful samples, and scans that were acquired during the Post Low stress and pre High/Low stress resting-state scans as negative.

## Appendix D. Voxel Normalization

The preprocessing step we call Voxel normalization can be thought of as z-scoring each voxel individually across the entire scan. it highlights the activation of a voxel relative to it's

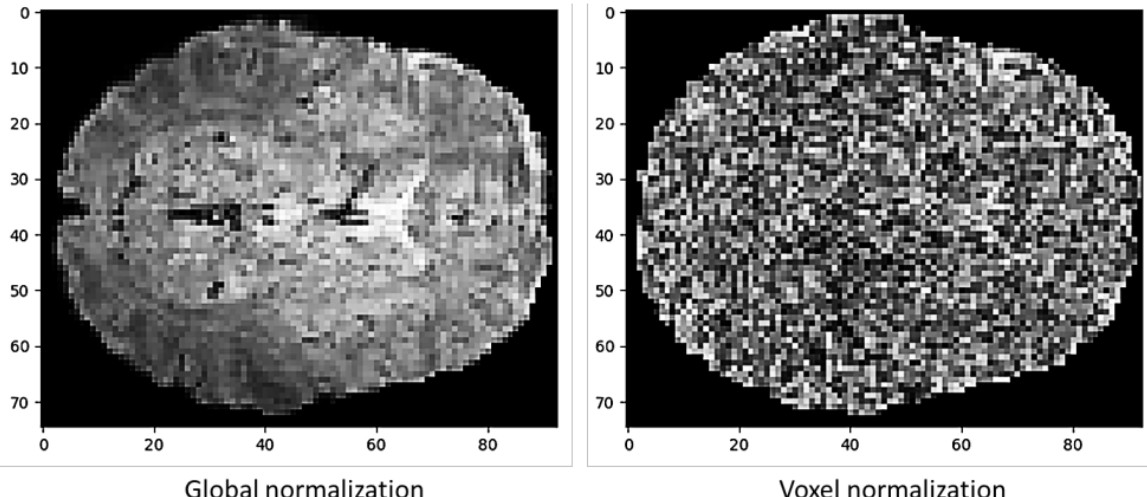

Figure 8: A comparison of global and voxel normalizations.

average activation throughout the scan. In contrast to Global normalization, that normalizes across all voxels, subtracting the global mean, and results in a structural enhancing image, the voxel norm enhances temporal information present in the bold signal.

## Appendix E. The differences between TFF and MetricFMRI

MetricFMRI introduces a pre-training that is based on fingerprinting and a novel metric learning approach for FMRI data. The method leverages triplets of anchor, positive and negative samples with a triplet loss objective. On the other hand, TFF is solely based on pre-training for reconstruction that can capture the variability in the data, but is not sufficient to emphasize the differences between different individuals.

Identifying personal factors in fMRI scans is mostly about the unique features of individuals with respect to other subjects. The proposed metric learning adheres to this principle. As shown in the ablation study, the novel triplet training phase is crucial for model performance.

The final model used in MetricFMRI and TFF during inference is composed of the same architecture, yet, the second pre-training phase in TFF is computationally expansive compared to the triplet training employed in MetricFMRI. Specifically, TFF employs an optimization for an encoder-transformer-decoder architecture, which requires 3-4 days of training on a fairly good GPU. In MetricFMRI, the triplet training operates solely on the encoder-transformer architecture, and it converges much faster (within 4-5 hours on similar hardware).

## Appendix F. More implementation details

All MetricFMRI models were trained on a single V100 GPU card, with 16GB memory. As we used a window size of $w = 30$, each sequence contained 30 fMRI frames. On our hardware, the propagation of a triplet of sequences (anchor, positive, and negative) reached

the maximal GPU memory usage and did not allow us to train with a larger batch size. Therefore, in this study, we did not experiment with hard mining, which was found beneficial in other metric learning approaches, and we leave this experiment to future research.

