# OpenReview forum: "Pre-Training Transformers for Fingerprinting to Improve Stress Prediction in fMRI"
_MIDL.io/2023/Conference — MIDL 2023 Oral_

### Official Review · Reviewer_jMB7 · 2023-02-03

**Confidence:** 5
**Preliminary Rating:** 5
**Recommendation:** Oral

**Summary:**

This manuscript presents MetricFMRI, a multi-stage network for efficient encoding and modeling of the functional MRI (fMRI) fingerprint. The authors map the findings to specific brain networks and visualize outcomes. A dedicated fMRI study into combat stress is adopted to demonstrate performance. The presented method is the only one obtaining significant performance, compared to adopted benchmark methods.

**Strengths:**

The strengths of the paper are the joint efficacy of pretraining and the use of a triplet loss for proximal mapping of same subject data. The efficiency of the network aids in obtaining good performance. An ablation study is performed to assess the efficacy of sub-components. This method may be readily applied in more studies.

**Weaknesses:**

None of the benchmark methods was trained with significant performance, which is surprising. Where they not designed for the dataset (size) at hand?

A discussion the results is lacking.

The visualization of outcomes is currently not part of the main paper.


**Deanonymize Review:**

yes

**Detailed Comments:**

Section 4.2 Here benchmark methods are presented in the Results section – please migrate to the Experiments part of the paper.

Typo's: sequence -> 8 degrees


**Paper Type:**

methodological development

**Questions To Address In The Rebuttal:**

1. A (brief) discussion on the ablation study is lacking; please interpret Table 3.

2. What was the size of the datasets that the reference methods were evaluated on? This study operates on typical modest-size data (n=50), and not on data of population studies. Please briefly discuss.

3. p6. The proposed MetricFMRI framework is claimed to be ‘much more efficient’. The computing time of TFF is reported to be large. Can the authors estimate the model sizes (#parameters?). This may be added to the text, or Table 1.

4. In terms of open science; will the code be released?

---

### Official Review · Reviewer_Yiej · 2023-02-03

**Confidence:** 3
**Preliminary Rating:** 3
**Recommendation:** Poster

**Summary:**

The authors proposed a novel method using 3D CNN and transformers to learn a fingerprint function in fMRI scans. Previous work focused on proposing metrics to identify fMRI fingerprints that belong to the same individual brain, and later works suggested that it’s possible to link the fingerprints with clinical and behavioral attributes. This work follows the stream of research on the latter topic. The proposed method consists of 3 steps, 1) reconstructing fMRI data using a 3D-CNN-based encoder and decoder to learn effective representations; 2) replacing the decoder with a transformer to enforce metrics in the representations; 3) fine-tuning the encoder+transformer network for a down-stream task in a supervised way.

**Strengths:**

The paper is overall well-structured and approaches an interesting problem, as to using machine learning methods to predict clinical attributes (stress prediction). The method is most similar to the transformer framework for fMRI (TFF) and improves it by adding a metric learning step for more representative embeddings of fMRI sequences. The method is supported by comparison against the baselines. The addition of triplet loss using transformers brings the most significant improvement.

**Weaknesses:**

There are some minor concerns about the architecture and evaluation:

- For the reconstruction phase, is it necessary to first train with a decoder and then replace it with a transformer instead of directly training with an encoder-transformer architecture for reconstruction?
- While TFF might be a less efficient architecture, the addition of the triplet loss in the work seems very effective for the task, is it possible to also add the triplet loss to TFF and observe it also improves its performance?
- In table 1 and table 2, what does BAC stand for?
- For the triplet loss, did the authors use hard example mining while training with triplet loss? Besides triplet loss, have the authors also considered other similar losses, such as contrastive loss?

**Deanonymize Review:**

no

**Paper Type:**

methodological development

**Questions To Address In The Rebuttal:**

- Justify the choice to first use a decoder and replace it with a transformer.
- Please explain the term "BAC", as well as compare with TFF with the triplet loss, if possible.
- Please provide some insights of using other metric loss.

---

### Official Review · Reviewer_ZrTu · 2023-02-03

**Confidence:** 3
**Preliminary Rating:** 3
**Recommendation:** Poster

**Summary:**

The authors introduce a method for fingerprinting on fMRI in order to improve stress prediction in fMRI. Fingerprinting produces unique representations for each subject. The authors deploy a transformer-based technology and pre-training and call the new tool MetricFRMI. Triplet training (anchor, positive, negative) is demonstrated to be key to improvements in fingerprinting.


**Strengths:**

Overall the submission is quite clear.

Detailed description of each component of the pipeline is included as well as an ablation study.

Explainability is included at the end and the preliminary data correlates well with literature findings.

**Weaknesses:**

The authors have described previous technical work related to stress prediction, but I missed a couple sentences about why that task is the one chosen as a test case here. Is stress prediction a highly challenging task or a well-fitting task to represent the capabilities of the new pipeline? I do not get a good sense how much of an impact any of the improvements described would bring to the field.

The submission has slight novelty. The submission introduces fingerprinting to the TFF (transformer framework for fmri) model. More details on how the adoption of TFF was done would be appreciated in order to better appreciate level / significance of contributions.

Performance metrics: It would have been welcome to briefly describe the performance metrics that are presented in Tables 1 and 2 and to fully discuss the contents of these. What is AUC vs AUROC? How is Acc defined? BAC results are not discussed.

**Deanonymize Review:**

no

**Detailed Comments:**

What is CLS? Classification / "collective learning systems"?

Much higher variance can be seen in Table 3 for the proposed full method. What is the explanation for that?

"vanilla" colloq


**Paper Type:**

both

**Questions To Address In The Rebuttal:**

Can you characterize how different your framework is from TFF?

Can you provide an intuition why stress prediction was chosen or why the proposed tool is applicable to this particular task?

I would have loved to see more of the explainability discussion in the main text as opposed to in the supplement.

---

### Meta-Review · Area_Chair_3pPb · 2023-02-24

**Recommendation:** Accept (Poster)
**Confidence:** 4

**Metareview:**

The authors introduce a method for fingerprinting (unique representation building) of fMRI to improve stress prediction using transformers.

The paper went through careful reviews and the authors responded with thorough discussion and modifications to the paper. I think that the reviewers had several confusions due to lack of clarify in the original submission, and the true impact of the paper is a bit still unclear.  I also think there are several promises still left to do, such as putting the code on github. I want to highlight that this was promissed during review, so must be done at camera ready deadline.

It does feel like the authors did address the major issues raised by reviewers, including substantial re-writes of parts (e.g. the introduction and discussion) as well as clarifications. Despite the focus on a specific task (stress), which reviewers highlighted, the dataset is larger than in many studies, which gives some additional support to the study. I am recommending acceptance, as it seems like a paper that merits presentation and discussion at the main conference.